# Learning to Exploit Stability for 3D Scene Parsing

**Yilun Du**
MIT CSAIL

**Zhijian Liu**
MIT CSAIL

**Hector Basevi**
University of Birmingham

**Aleš Leonardis**
University of Birmingham

**William T. Freeman**
MIT CSAIL

**Joshua B. Tenenbaum**
MIT CSAIL

**Jiajun Wu**
MIT CSAIL

## Abstract

Human scene understanding uses a variety of visual and non-visual cues to perform inference on object types, poses, and relations. Physics is a rich and universal cue that we exploit to enhance scene understanding. In this paper, we integrate the physical cue of stability into the learning process by looping in a physics engine into bottom-up recognition models, and apply it to the problem of 3D scene parsing. We first show that applying physics supervision to an existing scene understanding model increases performance, produces more stable predictions, and allows training to an equivalent performance level with fewer annotated training examples. We then present a novel architecture for 3D scene parsing named Prim R-CNN, learning to predict bounding boxes as well as their 3D size, translation, and rotation. With physics supervision, Prim R-CNN outperforms existing scene understanding approaches on this problem. Finally, we show that finetuning with physics supervision on unlabeled real images improves real domain transfer of models training on synthetic data.

## 1 Introduction

Human scene understanding is rich, and operates robustly using limited information. Physics comprises invisible causal relationships that are ubiquitous in natural scenes and crucial in scene understanding [Battaglia et al., 2013, Zhang et al., 2016]. In particular, the vast majority of natural scenes are *physically stable*, a prior most systems for visual scene understanding do not exploit.

Visual scene understanding takes many forms. Most commonly, elements of a scene are detected, classified, and localized, either through bounding boxes [Dai et al., 2016] or pixel labels [He et al., 2017]. If object instances are known, object poses can be inferred directly [Brachmann et al., 2016]. Supervision here can take the form of ground truth pixel annotations, as well as pixel depth if depth images are available. Physical supervision is more challenging to introduce because there are few visual features directly associated with physical relationships. Image sequences enable the robust inference of physical properties through movement of visual features [Stewart and Ermon, 2017], but analysis of single images requires a different approach.

Machine learning has enabled end-to-end inference of object physical properties [Wu et al., 2015], scene physical properties such as stability [Li et al., 2017], and prediction of future states [Lerer et al., 2016, Wu et al., 2017a]. However, these systems have been trained in heavily constrained domains and cannot easily be adapted to the variety present in natural scenes. In contrast, systems that incorporate physics engines operate on discrete, interpretable representations; they are amenable to levels of object abstraction, such as the approximation of geometry by simple geometric primitives, and are more easily adaptable. Further, these types of abstractions can be more readily extracted from natural scenes, as demonstrated by existing machine learning systems for semantic localization

---

Correspondence to: jiajunwu@mit.edu

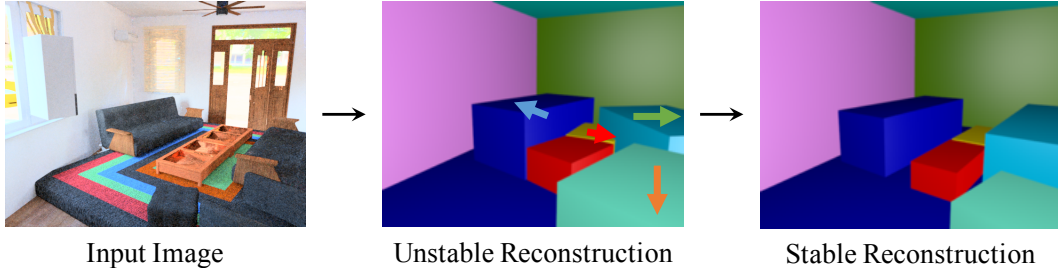

| Input Image | Unstable Reconstruction | Stable Reconstruction |

Figure 1: When we observe natural scenes, we understand that objects are physically stable. We wish to harness this inherent stability signal to generate modifications to model predictions that are physics stable. In the scene above, stability signals allow us to modify predictions due to object intersection (red, green and blue arrows) and due to bad alignment with walls (orange arrow).

and segmentation [Song et al., 2017, He et al., 2017], and facilitate combinations of systems for understanding of natural scenes and physics engine–based stability supervision.

Physical stability supervision is not used by modern scene reconstruction systems [Tulsiani et al., 2018], but can have multiple benefits as seen in Figure 1. First, it can correct certain types of pose errors resulting from the limited resolution of visual features. These errors manifest in objects intersecting other objects, objects intersecting the boundaries of the scene, and objects not aligned with their supporting surfaces. When abstracted to simple geometric primitives, these errors can be identified by a physics engine with high accuracy and efficiency, and corrected to both increase accuracy and the *physical stability* of the reconstructed scene.

Second, physical stability supervision is always applicable and requires no annotation. This enables the inclusion of unlabeled data in the learning, which facilitates the training of systems that generalize to natural scenes. This is particularly important due to the impracticality of annotating sufficient data to cover the variety of visual features and spatial configurations present in natural scenes.

In this paper, we systematically explore how the use of scene stability as a supervisory signal can enhance scene understanding by improving the quality of solutions and reducing the need for annotated training data. We apply this to the problem of scene reconstruction, which consists of identifying objects present in a scene, generating the 3-dimensional bounding box and 4-dimensional pose for each object. We train using synthetic data for which ground truth bounding box annotations are available, and also take advantage of data from real scenes without annotations by using scene stability as a supervisory signal, under the natural assumption that real scenes are stable. We incorporate stability supervision via the use of a physics engine, and estimate gradients using REINFORCE [Williams, 1992].

We evaluate our framework across several photosynthetic and realistic domains: human-designed room layouts from SUNCG [Song et al., 2017], photo-realistically rendered automatically generated room layouts from SceneNet-RGBD [McCormac et al., 2017], and real scenes from SUN-RGBD [Song et al., 2015]. We validate that our framework makes use of unlabeled data to increase reconstruction performance and demonstrate that with physics supervision, we require fewer annotations to achieve the same performance as a fully-supervised framework.

Our contributions are three-fold. First, we propose a framework to integrate physical stability into scene reconstruction and demonstrate its ability to improve data efficiency and its use of unsupervised data. Second, we propose an end-to-end scene reconstruction network that achieves state-of-the-art performance on SUNCG [Song et al., 2017]. Third, we show our framework helps models trained on synthetic data to transfer to real data.

## 2 Related Work

Physical scene understanding has attracted increasing attention in recent years [Jia et al., 2013, Zheng et al., 2013, Shao et al., 2014, Zheng et al., 2015, Fragkiadaki et al., 2016, Finn et al., 2016, Battaglia et al., 2016, Chang et al., 2017, Li et al., 2017, Ehrhardt et al., 2017]. Beyond answering "what is where", physical scene understanding models the future dynamics of objects [Lerer et al., 2016, Mottaghi et al., 2016] and facilitates inference of actions to reach a goal [Li et al., 2017]. Many previous papers have inferred physics from pixel-level representations [Lerer et al., 2016, Mottaghi et al., 2016, Wu et al., 2016, Wu, 2016]; a number of methods infer physics from voxel-level

representations [Zheng et al., 2013, 2015, Liu et al., 2018]; others propose to learn a flexible model of object interactions, but assume an existing decomposition of a scene into physical objects [Chang et al., 2017, Battaglia et al., 2016, Fragkiadaki et al., 2016]. Our framework combines a flexible model of object interactions with the complex task of inferring the layout and object locations and poses in natural indoor scenes.

Our work integrates physics and geometric context into solutions to the task of 3D scene reconstruction, which was first presented in Robert's blocks world [Roberts, 1963], and repeatedly revisited in later studies with various techniques [Gupta et al., 2010, Silberman et al., 2012, Hoiem et al., 2005]. Recent work in 3D scene reconstruction has explored reconstruction representations such as depth [Eigen and Fergus, 2015], surface normals [Bansal and Russell, 2016], and volumetric reconstructions [Firman et al., 2016, Song et al., 2017]. Our work builds on the explicit object-based geometric primitive representation of 3D scenes in Tulsiani et al. [2018], which facilitates physical prediction.

Previous approaches applying complex physics to 3D scene reconstruction have focused on inferring occluded portions of objects. Zheng et al. [2013, 2015] have applied stability constraints to segment point clouds into grouped primitives. The stability-based completion of objects assumes that occluded portions of objects extend to the nearest boundary, and does not generalize in cases when such assumptions do not hold. The approach of Shao et al. [2014] uses stability as a selection criterion to infer occluded portions of objects in the form of geometric primitives. Their approach uses fixed heuristics to infer cuboids to add to an existing partial object shape but suffers when two objects are close to each other. The approach of Jia et al. [2013] constructs geometric primitives for objects by fitting surfaces one-by-one to depth information, and applies simple rules about supporting surfaces and center of mass to approximate features related to stability. The approach of Gupta et al. [2010] constructs physical representations of objects but uses a set of heuristic rules to infer physics. By leveraging the representation of objects as geometric primitives consistently in our scene reconstruction and stability estimation components, our model is able to handle partially occluded objects (Figure 6) and nearby objects and operate on the diversity of scene configurations and objects found in natural scenes. The representation of objects as geometric primitives couples naturally with the representations employed by the physics engine that we use to estimate scene stability, and the high fidelity of the physics engine provides more accurate and direct physics supervision than used in other works.

Our results relate to research that uses deep networks to explain scenes with multiple objects [Ba et al., 2015, Huang and Murphy, 2015, Eslami et al., 2016, Wu et al., 2017b]. A few recent studies have also modeled higher level relations in scenes [Fisher et al., 2011, 2012]. Our work extends the model introduced by Tulsiani et al. [2018], which inverts the geometric shapes and scene layout from a single image, but without modeling physics.

## 3 Physics Stability Model

To recover a 3D reconstructed scene, our physics stability model consists of (i) a primitive prediction module—an inverse graphics component to build object representations from an input image, (b) a layout prediction module that estimates the enclosing space of objects, and (c) a physics stability module that simulates the stability of the prediction. We show our framework in Figure 2.

### 3.1 Overview

The first component of our model is an inverse graphics component to estimate the physical state of all objects in an image. We represent each object present in a image as a 3D bounding box primitive and predict translation, scale and rotation of the primitive in full 3D space. These primitive predictions are generated through two different architectures, one which generates 3D primitives through 2D bounding boxes and another which predicts 3D primitives in parallel with 2D bounding boxes. The second component of our model is a layout prediction module that represents the layout around all primitives as a set of normal planes. The third component of our model is a physics stability module that takes predictions from both primitives and layout to generate a 3D scene and infer the stability of each of the primitives through a physics engine [Coumans, 2010].

Our model is able to combine powerful neural networks for primitive and layout prediction with a real world physics engine. This allows us to generate predictions that are interpretable and feasible in the real world. Our model can be trained in an semi-supervised manner on images without 3D annotations by learning to generate predictions that are physically stable.

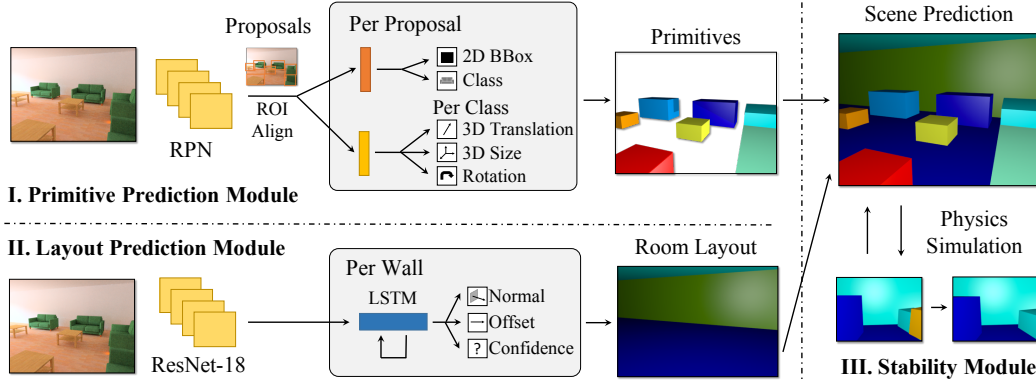

Figure 2: **Overview of the Physical Stability Model (PSM)**. Our model consists of three modules. A primitive prediction module predicts each object in a scene as a set of cuboid primitives, a layout prediction module predicts the walls surrounding a scene, and a physics stability module provides feedback on the stability of the predictions. We show our end-to-end primitive prediction network (Prim R-CNN) for the primitive prediction module; please see Tulsiani et al. [2018] for their Factored3D model.

## 3.2 Primitive Prediction Module

To show generality of our physics supervision, we consider two different primitive prediction models.

**Ground Truth Bounding Boxes** Given that we have access to ground truth bounding boxes of objects in an image, we use the Factored3D architecture described in Tulsiani et al. [2018]. The Factored3D architecture encodes images through both a coarse and fine ResNet-18 encoder [He et al., 2015] and then uses ROI pooling to extract relevant image features. These features are concatenated with ground truth bounding box coordinates to regress parameters for size, translation and classify rotations for each primitive.

**Raw Input Image** For direct primitive prediction from a raw image, we propose a new network architecture called Prim R-CNN. We use a Faster R-CNN [Ren et al., 2015] architecture with a ResNet-50 Feature Pyramid Network (FPN) [Lin et al., 2017] to extract features from images. The Faster R-CNN architecture consists of a region proposal network (RPN) to propose candidate bounding boxes and a R-CNN network to refine bounding boxes and classify their content. Inspired by the Mask R-CNN architecture [He et al., 2017], in parallel to bounding box prediction and class refinement, we add a primitive prediction branch that independently regresses size, translation and classifies rotation for each possible class for each candidate bounding box, allowing priors to be formed for each class type. We use 24 rotation bins as in Tulsiani et al. [2018]. An overview of our network architecture can be found in Figure 2 with architecture details in the supplementary material. In comparison, the Factored3D model generalizes to raw images by first using a off-the-shelf bounding box detector Edge Boxes [Zitnick and Dollár, 2014] to propose bounding boxes and then by forwarding each resultant proposal through a Factored3D model with an added head for existence classification.

Our overall architecture is end-to-end and allows features learned for 2D detection to also be used in primitive prediction. Furthermore, we incur reduced computational cost during training time as we are able to offload much of initial bounding box computation to the RPN as opposed to forwarding all possible candidates through the Factored3D model.

**Prim R-CNN Loss Function** To train our network, we use the loss function $\mathcal{L} = \mathcal{L}_{\text{cls}} + \mathcal{L}_{\text{box}} + \mathcal{L}_{\text{size}} + \mathcal{L}_{\text{trans}} + \mathcal{L}_{\text{rot}}$. We use $\mathcal{L}_{\text{cls}}$ and $\mathcal{L}_{\text{box}}$ to represent the losses for classification and bounding box regression respectively as defined in Ren et al. [2015]. Given ground truth labels for translation, $t$, and size, $s$, and model predictions of $\hat{t}$ and $\hat{s}$, we have $\mathcal{L}_{\text{trans}} = \|t - \hat{t}\|^2$ and $\mathcal{L}_{\text{size}} = \|s - \hat{s}\|^2$. We define the rotation loss $L_{rot}$ assuming a predicted rotation distribution of $k_d$ and ground truth rotation bin $g$ as $\mathcal{L}_{\text{rot}} = -\log(k_d(g))$. We only apply the above losses to primitives whose corresponding predicted bounding boxes have significant overlap with ground truth bounding boxes.

## 3.3 Layout Prediction Module

A second component of our model is the layout prediction module. We represent the layout around an image as a set of walls, floor and ceiling. We represent each such surface as a plane, parametrized as the set of points $\mathbf{x} \in S$ where $S = \{\mathbf{x} | \mathbf{n} \cdot \mathbf{x} + d_0 = 0\}$

We predict each plane by predicting a normal vector $\mathbf{n}$, with $\|\mathbf{n}\| = 1$, and a distance offset $d_0$. Our layout prediction network consists of an LSTM with three outputs at each time step, consisting of a plane existence probability, $\mathbf{n}$, and $d_0$. We initialize the hidden state and input into the recurrent network as a convolutional encoding of the scene. Planes, if they exist, are predicted in the order floor, ceiling, walls from left to right. We use $\mathcal{L}_2$ loss on normal and offset predictions and cross entropy loss on existence probabilities. Details about network architecture can be found in the supplement.

## 3.4 Physics Stability Module

Throughout the paper, we use a rigid body physics simulator, Bullet [Coumans, 2010], for estimating 3D object stability. Since the simulator is not differentiable, we train both layout prediction and primitive prediction modules jointly using a stability signal from multi-sample REINFORCE [Williams, 1992]. Our physics stability module operates on primitive cuboid representations of objects. We found that representing objects as voxels led to the same approximate performance but required significantly more expensive 3D simulation.

**Stability Calculation**    Given predictions from both our layout and object prediction modules, we infer the direction of gravity from the floor prediction. We then initialize all predicted oriented bounding boxes in Bullet with the same mass and friction coefficient. We simulate all objects for 50 seconds to detect even small instabilities. Each primitive then receives binary stability labels of 0 and 1 dependent on object displacement. The overall stability score of scene $S(c)$ is calculated by taking the average of stability scores of all primitives, so in the situation where a scene has 1 stable object and 1 unstable object, the calculated stability score would be 0.5.

**REINFORCE Details**    We train layout and primitive prediction modules simultaneously with the stability signal and apply REINFORCE at the scene level. We vary predictions from both modules simultaneously and sample translations, scales, and offsets of predicted wall planes from a normal distribution and size from a log-normal distribution.

Given sampled values for primitives and layouts in a configuration $c$, we compute the overall probability of the configuration $P(c)$ as well as the corresponding stability score $S(c)$. Our overall loss for reinforce is then $\mathcal{L}_{\text{stab}} = -\sum_c \log(P(c)) \left(S(c) - \overline{S(C)}\right)$, where we subtract the average stability $\overline{S(C)}$ across different sampled configurations from each individual calculated stability score. We sample a total of 15 different sets of primitive proposals for each scene.

# 4 Evaluation

We evaluate our physics model on three different scenarios: synthetic room images from both the SUNCG dataset [Song et al., 2017] and SceneNet RGB-D dataset [McCormac et al., 2017] and real images from the SUN RGB-D dataset [Song et al., 2015]. We show that our physics-based supervision helps performance in both uncluttered realistic room scenes in the SUNCG dataset and cluttered scenes in the SceneNet RGB-D dataset. We show that our physics stability model can take advantage of not only synthetic data, but real data without 3D annotations, hinting that physics can help model transfer from synthetic to real data.

## 4.1 Scene Parsing with Ground Truth 2D Bounding Boxes

We begin by showing that our proposed physical stability model provides gains to performance of the Factored3D model using ground truth 2D bounding boxes. It helps by more effectively using labeled data and further data without 3D annotations.

**Data**    We evaluate on the SUNCG dataset [Song et al., 2017] and SceneNet RGB-D [McCormac et al., 2017] synthetic datasets. We use physically based renderings for SUNCG found in Zhang et al. [2017]. We use splits of the SUNCG dataset in Tulsiani et al. [2018] with around 400,000 training images, 50,000 validation images and 100,000 test images. The set of objects in SUNCG is diverse, including lights, doors, candlesticks; predicting the 3D location all such objects is beyond the scope of the work. We restrict our predictions to beds, sofas, chairs, refrigerators, bathtubs, tables, and desks.

| Dataset | # Labels | Model | $S_{\mathrm{Phys}}$ | $\mathrm{mAP}_{0.5}$ | $\mathrm{mAP}_{0.3}$ | $\mathrm{mAP}_{0.1}$ | Avg IoU |
|---|---|---|---|---|---|---|---|
| SUNCG | 0.1% | Factored3D | 0.23 | 0.048 | 0.280 | 0.623 | 0.149 |
| | | Factored3D + **P** | 0.28 | 0.067 | 0.335 | 0.671 | 0.164 |
| | | Factored3D + **P** + **FT** | 0.41 | 0.076 | 0.331 | 0.650 | **0.164** |
| | 1% | Factored3D | 0.36 | 0.100 | 0.389 | 0.713 | 0.200 |
| | | Factored3D + **P** | 0.38 | 0.122 | 0.437 | 0.743 | 0.206 |
| | | Factored3D + **P** + **FT** | 0.54 | 0.124 | 0.450 | 0.751 | **0.208** |
| | 10% | Factored3D | 0.31 | 0.202 | 0.540 | 0.809 | 0.256 |
| | | Factored3D + **P** | 0.54 | 0.213 | 0.554 | 0.813 | 0.261 |
| | | Factored3D + **P** + **FT** | 0.59 | 0.212 | 0.557 | 0.812 | **0.266** |
| SceneNet RGB-D | 100% | Factored3D | 0.21 | 0.337 | 0.091 | 0.010 | 0.0276 |
| | | Factored3D + **P** | 0.27 | 0.349 | 0.089 | 0.010 | **0.0304** |

Table 1: Quantitative results of Factored3D trained with limited data on SUNCG. **P** represents physics supervision while **FT** represents finetuning. **FT** and **P** both improve performance.

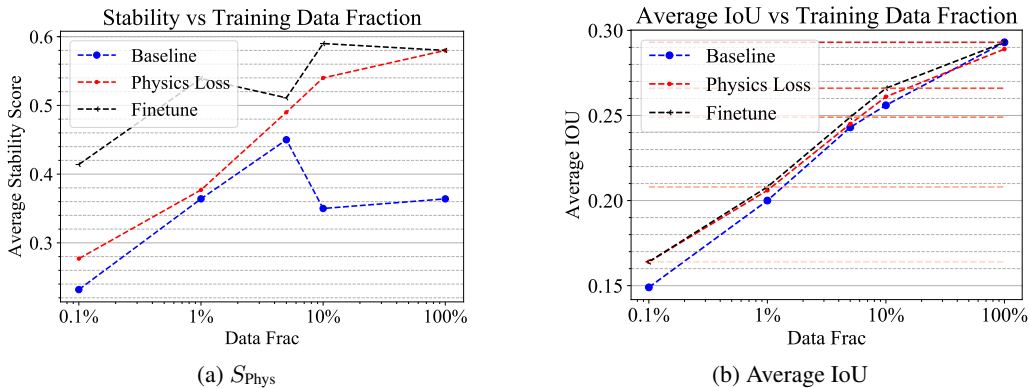

(a) $S_{\mathrm{Phys}}$
(b) Average IoU

Figure 3: Average IoU vs data fraction used to trained Factored3D model on SUNCG. Finetuning with physics always gives around a 50% data efficiency boost and improves stability.

We also evaluate on the SceneNet RGB-D dataset to show that our physics-based method also generalizes to very cluttered scenes. Primitives in SceneNet RGB-D are dropped via simulation using a physics engine and are often stacked. We split the SceneNet RGB-D dataset into 90% training and 10% testing. We use a subsection of the SceneNet RGB-D dataset of approximately 300,000 images.

**Setup** We test how our physics stability model can be used to take fuller advantage of both limited supervised data and data without 3D annotations.

Our training protocol consists of three different steps. We first train both our primitive prediction module and layout prediction module using existing labeled data. Second, we train both modules with the addition of physical stability module. We find that adding the physical stability module before pretraining leads to slow training, possibly due to there being many possible stable positions that are far away from ground truth. Third, we finetune using our remaining semi-supervised data without 3D annotations (so in the case of 1% data use, we use 99% of data without 3D annotations), containing only color images and ground truth bounding box annotations, to train our model using the physics stability module by using alternate batches of supervised and semi-supervised data. We use the same training settings for Factored3D as described in Tulsiani et al. [2018].

To quantify our results, we compute intersection over union (IoU) values between each predicted primitive from a ground truth bounding box and the corresponding ground truth primitive label. We use IoU as opposed to thresholds for closeness values in Tulsiani et al. [2018] as they more accurately represent how close a predicted primitive is to a real primitive. To further quantify our results, we also compute IoU values between each predicted primitive and all possible ground truth primitives in a scene. We then compute mean average precision (mAP) values for ground truth primitives assuming thresholds for IoU matching of 0.1, 0.3, and 0.5.

| Model | Walls | Stability | 0.1 mAP | 0.3 mAP | 0.5 mAP |
|---|---|---|---|---|---|
| Factored 3D (GT Boxes) | None | 0.34 | **0.851** | 0.623 | 0.308 |
| Factored 3D (Edge Boxes) | None | 0.10 | 0.790 | 0.296 | 0.046 |
| Prim RCNN | None | 0.42 | 0.808 | 0.672 | **0.402** |
| Prim RCNN+Physics | Predicted | **0.54** | 0.814 | **0.680** | 0.393 |

Table 2: Quantitative results for training on the full SUNCG dataset. Prim R-CNN achieves better performance than Factored3D Models. Physics offers additional benefits.

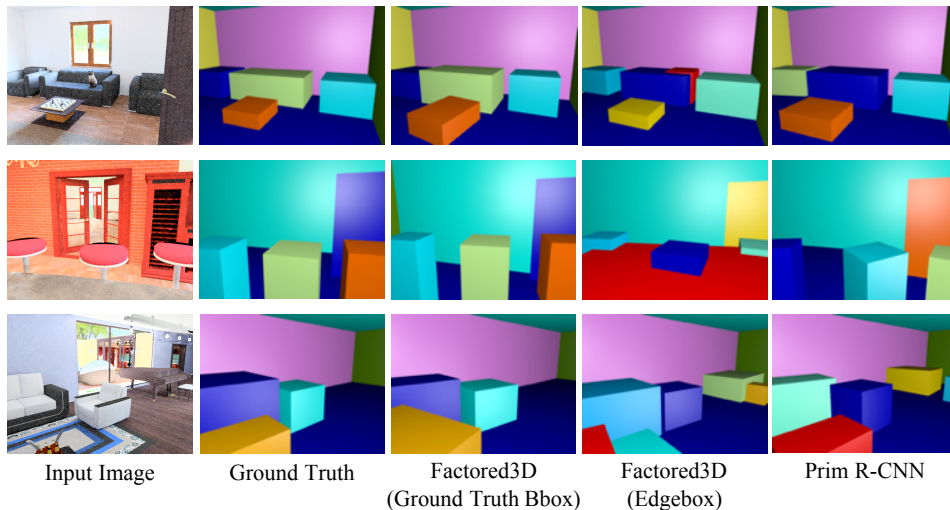

| Input Image | Ground Truth | Factored3D (Ground Truth Bbox) | Factored3D (Edgebox) | Prim R-CNN |

Figure 4: Qualitative results on SUNCG dataset. Prim R-CNN is able to detect objects more reliably and infer localization better then Factored3D with Edge Boxes.

**Results**    We find that our physics stability module significantly improves the stability of network predictions and also improves overall primitive prediction metrics. We present quantitative results for training each of the three steps of our models in Table 1 and a plot of trend in performance with different data fractions in Figure 3.

Our full model is able to achieve 50% increased data efficiency at a wide range of fractions of data usage on the SUNCG dataset with semi-supervised finetuning. Our model is also able to construct scenes with much higher physical stability after training. We found diminishing gains from the physics module on the whole SUNCG dataset (approximately 400,000 images), perhaps because on a scale of so many images, primitives are already predicted at a close-to-stable location, reducing need of physics supervision, although we still observe increased stability. We find that at full dataset scale, physics provides minor improvements to layout prediction, reducing wall offset prediction MSE from 0.299 to 0.297, while surface normal MSE stays constant at 0.013.

We further test our physics loss on the SceneNet RGB-D dataset and are also able to observe gains on the full dataset as seen in Table 1. This indicates that our stability supervision is also applicable in cases of very cluttered scenes.

## 4.2    Scene Parsing from Raw Images

We further show that our proposed physical stability model is able to provide gains to an end-to-end primitive prediction network with further gains from utilizing unlabeled input images. We also show that our proposed model outperforms the previous state-of-the-art method (Factored3D).

**Setup**    We use the same data and training process as that described in Section 4.1. The only exception is when undergoing stability training, since each predicted primitive now has an associated confidence score, when we simulate a scene of primitives, we now simulate sets of primitives predicted above sets of threshold confidence scores.

For metrics, since Prim R-CNN predicts possible sets of bounding boxes, there is no longer a direct correspondence between predicted primitives and ground truth primitives. Therefore, we cannot

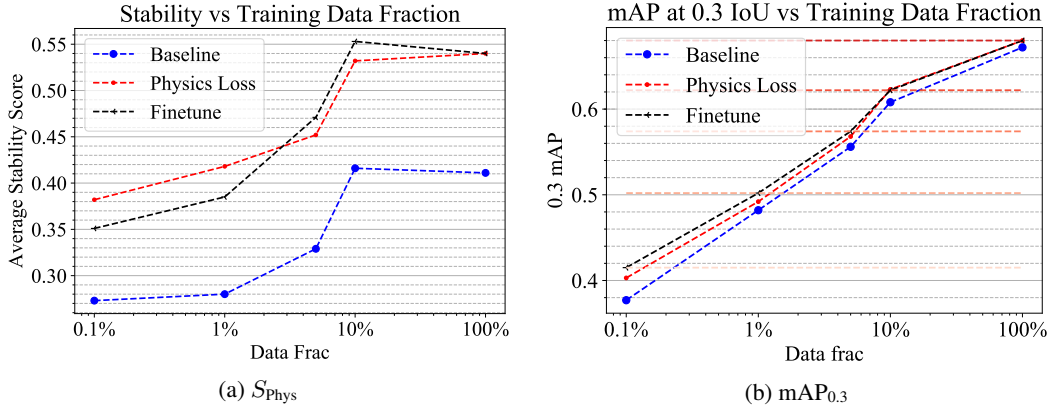

(a) $S_{\text{Phys}}$
(b) $\text{mAP}_{0.3}$

Figure 5: Plots of Prim R-CNN's performance using limited data on SUNCG. Finetuning and physics improve performance, providing over 200% data efficiency at 0.1% data use.

| # Labels | Model | $S_{\text{Phys}}$ | $\text{mAP}_{0.5}$ | $\text{mAP}_{0.3}$ | $\text{mAP}_{0.1}$ |
|---|---|---|---|---|---|
| 0.1% | Factored3D | 0.27 | 0.105 | 0.377 | 0.629 |
| | Factored3D + **P** | 0.38 | 0.117 | 0.403 | 0.627 |
| | Factored3D + **P** + **FT** | 0.35 | 0.115 | **0.415** | 0.636 |
| 1% | Factored3D | 0.28 | 0.191 | 0.482 | 0.710 |
| | Factored3D + **P** | 0.42 | 0.195 | 0.492 | 0.704 |
| | Factored3D + **P** + **FT** | 0.39 | 0.187 | **0.502** | 0.729 |
| 10% | Factored3D | 0.416 | 0.303 | 0.608 | 0.796 |
| | Factored3D + **P** | 0.53 | 0.300 | 0.623 | 0.802 |
| | Factored3D + **P** + **FT** | 0.55 | 0.300 | **0.622** | 0.805 |

Table 3: Quantitative results on training Prim R-CNN on the SUNCG dataset. **P** represents physics supervision, **FT** represents finetuning. **P** and **FT** help mAP metrics and stability.

compute the average IOU metric used in Section 4.1 and only report the mAP scores defined in Section 4.1.

**Results**　We show quantitative results for training Prim R-CNN in Table 2. Prim R-CNN out performs both Factored3d trained with or without ground truth bounding boxes, using similar number of parameters and faster training time. Our model achieves further gains from physics when trained on all data in SUNCG. We show qualitative results on the SUNCG dataset in Figure 4.

We show plots for Prim R-CNN trained and finetuned with physics on limited data in Figure 5 and quantitative numbers in Table 3. We note that our model achieves 200% data efficiency at 0.1% data utilization with over 50% data efficiency at 10% data utilization. We further note that with physics supervision, we get the same performance at 0.3 IOU mAP as the ground truth Factored3D model using only 10% of the data.

## 4.3　Real Data Transfer

To demonstrate the generalization of our approach, we further show that our model can be finetuned on real images and obtain significant gains to performance on SUN RGB-D without using **any** 3D labeled real data.

**Data**　We use 10,335 real training color images from the SUN RGB-D [Song et al., 2015] with 2D bounding box annotation.

**Setup**　Our training protocol consists of two steps. We first fully train both a primitive prediction module and a layout prediction module on SUNCG. Next, we finetune these modules on real images from SUN RGB-D, by mixing a batch of labeled RGB data from SUNCG with a unlabeled raw images from SUN RGB-D labeled with only our physics stability module. In the case of Prim R-CNN, when finetuning on unlabeled raw images from SUN RGB-D, we also train the corresponding bounding box classification parts of the network. We use the mAP metrics in Section 4.1.

| Model | Finetune | Stability | 0.1 mAP | 0.3 mAP | 0.5 mAP |
|---|---|---|---|---|---|
| Factored3D (GT Boxes) | − | 0.23 | 0.238 | 0.016 | 0.000 |
| Factored3D (GT Boxes) | + | **0.49** | **0.367** | **0.056** | **0.012** |
| Prim R-CNN | − | 0.07 | 0.193 | 0.024 | 0.000 |
| Prim R-CNN | + | **0.22** | **0.276** | **0.056** | **0.060** |

Table 4: Quantitative result on finetuning on the SUN RGB-D Dataset. We note that finetuning leads to large increases on performance in both Factored3D and Prim R-CNN.

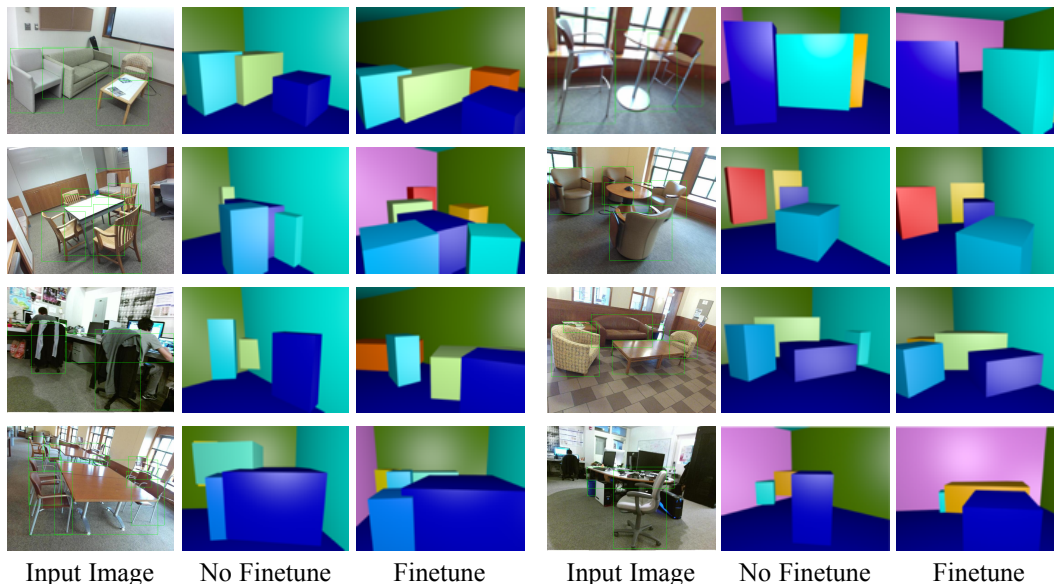

| Input Image | No Finetune | Finetune | Input Image | No Finetune | Finetune |

Figure 6: Qualitative Results on the SUN RGB-D dataset. Finetuning places predicted primitives on the floor and reduces collisions.

**Results**   We show quantitative results of finetuning on SUN RGB-D in Table 4 and qualitative results of finetuning in Figure 6. Our model has significant improvement in performance on metrics after finetuning. Qualitatively, we observe finetuning allows spread out configurations of objects, lowers primitives to the ground, and pushes back walls.

Quantitatively, our numbers are relatively low. We suspect that one reason is sensor disparity between images in SUNCG and SUN RGB-D. From a qualitative point of view, it appears our overall model is capable of making more realistic looking reconstructions of the original input image.

**Human Results**   We further evaluate our results on humans to evaluate if finetuning makes scenes more qualitatively reasonable to humans. We randomly sample 100 images from SUN RGB-D and construct 3D reconstructions using both a finetuned and non-finetuned Factored3D model. We evaluate each image with 20 different people through Amazon Mechanical Turk. We found that in 13.3% of scenes, people disliked the finetuned Factored3D result (<35% approval), in 52.4% of scenes people were indifferent (between 35% approval and 65% approval) and in 32.5% of scenes peoples preferred reconstructions from the finetuned model (>65% approval). Overall, we find that many scenes are more appealing to humans after finetuning. We also note a significant number of scenes that humans are indifferent towards—these may be due to the scenes already being stable or both reconstructions being of sufficiently poor quality that humans are indifferent. We include finetuned scenes that are most and least liked by people in the appendix.

## 5   Conclusion

We propose a physical stability model that combines primitive and layout prediction modules with a physics simulation engine. Our model achieves state-of-the-art results on the SUNCG dataset. Our model effectively uses unlabeled images for training and allows better domain transfer when applied to real images. We expect our model to have wider impact in the future due to the growing need for accurate 3D scene reconstruction methods and the increased prevalence of synthetic datasets.

**Acknowledgements** This work is in part supported by ONR MURI N00014-16-1-2007, the Center for Brain, Minds, and Machines (CBMM), Toyota Research Institute, NSF #1447476, and Facebook. We acknowledge MoD/Dstl and EPSRC for providing the grant to support the UK academics' involvement in a Department of Defense funded MURI project through EPSRC grant EP/N019415/1.

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
