[Supplementary Material]

## A.1 Examples of Human AMT Preferences

Figure A1 shows example 3D reconstructions most and least preferred by human AMT workers. In not preferred scenes, neither reconstruction is physically reasonable, while in preferred scenes, the fine-tuned scene is significantly more physically reasonable.

Figure A1: Panels showing fine-tuned scenes that are most and least preferred by AMT subjects when compared to scenes without fine-tuning. Fine-tuning has a notable impact on the best panels; on the worst panels, both visualizations look physically unreasonable.

## A.2 Network Architectures

**Layout Network Architecture** For the layout network, we take a $128 \times 256$ images and a $480 \times 640$ image and apply two different ResNet-18 networks on the two separate scales images. We encode each image using the first 4 blocks of ResNet-18. We then take the max pool of both encoding to get a 1,024 dimensional convolutional encoding of the image which we call $z$.

We take $z$ and apply a fully connected network to get a 512-dimensional encoding that we use to initialize the cell and hidden state of a 256-dimensional LSTM. We further also take $z$ and apply a fully connected network to get a 256-dimensional input feature $x$ to our LSTM, in addition to a start token.

We continuously feed $x$ into the LSTM network, and map the changing hidden state to the predict normal, distance and existence probability of a wall, which we call $o$. For inputs that are not the first input, we concatenate $o$ to $x$ and feed into the LSTM network.

**Prim R-CNN Architecture** All portions of Prim R-CNN correspond to portions of Faster R-CNN except for the primitive branch. In parallel to the ROI Align Pooling done to predict bounding boxes and classes as done in Faster R-CNN, we apply another ROI Align Pooling with a $14 \times 14$ feature map. We apply 4 sequential convolutional filters each with output channels equal to 256 and a kernel size of 3. We then map the 256 channels to channels equal to the number of classes. We flatten each of the resultant features and apply 3 different fully connected networks to regress rotation, size, and translation.