[Reviews · NeurIPS 2018]

Reviewer 1



The goal of this paper is to output a set of 3D bounding boxes and set of dominant planes for a scene depicted in a single image. The key insight is to incorporate stability constraints in the 3D layout, i.e., the reconstructed 3D boxes should not move too far under simulation (in Bullet) with physical forces (gravity, friction). Parameters for 3D boxes are regressed using a modified R-CNN training loss and dominant planes for the walls and floors are regressed via a RNN. A stability criterion is used to update the output 3D scene (via REINFORCE) where the predicted 3D layout is run through Bullet simulator and 3D displacements are checked. Results are shown on synthetic (SUNCG, SceneNet RGB-D) and real (SUN RGB-D) datasets, out-performing the factored 3D approach of [Tulsiani18]. Positives: I like the direction this paper takes towards incorporating physics for this problem. The paper demonstrates an interesting approach that leverages modern recognition components and shows positive results on a hard problem. I also like the analysis performed on synthetic and transfer to real scenes. The paper is well written (particularly given the amount of conveyed information in tight space) and reasonably cites prior work (although I have a couple more reference suggestions below). Negatives: My comments are mostly minor. A. Probably the biggest issue is the submission missed a highly relevant paper that also used stability constraints for inferring 3D scene layout: Abhinav Gupta, Alexei A. Efros and M. Hebert, Blocks World Revisited: Image Understanding Using Qualitative Geometry and Mechanics. In ECCV 2010. There are big differences between the current submission and the above paper ([Gupta10] infers layout for outdoor scenes, have a different mechanism for reasoning about stability, and use 2010 pre-deep learning tech). I think it would be good to tone down a bit the first contribution in the introduction in light of this paper. B. The second contribution mentions that “a end to end [sic x2] scene reconstruction network” is proposed. Since the simulator is not differentiable (L172), is this still considered an end-to-end network? In general, I think this would be a good poster. I like the overall story and the execution looks good. There are many system components, so I think it would be hard to reproduce without source code. Minor comments: + L161 “parameterizated” + Why is there a big bend around ~5x10^-2 in Fig 3(a) and at 10^-1 in Fig 5(a)? + The paper mentions that the model is trained with unlabeled synthetic data. However, it looks like ground truth bounding box annotations are used (L211). If this is the case, it would be great to rephrase this to avoid confusion since it’s not really unlabeled. + It may be good to also include some relevant work on higher-order 3D scene layout modeling, e.g., Characterizing Structural Relationships in Scenes Using Graph Kernels. Matthew Fisher, Manolis Savva, Pat Hanrahan. SIGGRAPH 2011.

Reviewer 2



Update: Based on the rebuttal I tends to accept this paper. The rebuttal solves most of my concerns. Although the quantitative improvements of using physics reasoning is not impressive as stated by R3, I believe incorporating physics reasoning is an inspiring approach to the community of 3D scene understanding. It's interesting to know voxel representation could not help much in stability reasoning compared with cuboid representation. Therefore I'd suggest the authors to include the clarifications in the rebuttal in their final draft and push forward a quicker code release for reproducibility. This paper proposes Prim RCNN, which involves stability reasoning for 3D objects and layout estimation in single RGB image of an indoor scene. The novelty point is incorporating physics reasoning in neural network which can be trained end-to-end. However my major concern is the missing comparison with state-of-the-art 3D scene estimation approach to validate the contribution of stability reasoning: 1) Although Table 2 shows better performance of Prim-RCNN with stability reasoning, it's not clear to me whether a good 3D object detection without stability reasoning can achieve better performance. I suggest the author to compare with the state-of-the-art 3D scene recovery approach: IM2CAD (Izadinia, Hamid, Qi Shan, and Steven M. Seitz, CVPR 2017 ) to validate the Prim RCNN approach. IM2CAD also predicts on both layout and shapes from single RGB image based on a Faster-RCNN structure, with detailed CAD model representation rather than 3D bounding boxes (which I believe is better). Unfortunately the author missed citing this paper. 2) Since SUNRGBD dataset has groundtruth 3D bounding box labeling, it's intuitive to directly train and evaluate on the SUNRGBD dataset. Thus the contribution of transferring from synthetic to real is not clear to me. 3) The User study in the experiments section is not convincing. Since the approach produces 3D bounding box of shapes, it tends to be "3D detection" rather than "3D reconstruction". In this case it's hard for users to say which is more physically reasonable by observing 2D images with rendered 3D blocks (often occluded) from single view only. 4) Quantitative evaluation on the layout estimation with or without physics is missing. 5) The proposed method is built upon Factored 3D, by substituting the detailed voxel-based shape prediction branch to simple 3D bounding box detection. However I believe predicting on detailed shapes (e.g. CAD models or voxel based representation) and layouts is more suitable for physical reasoning. Therefore the author's approach tends to be a preliminary exploration to involve stability reasoning in the neural network design. Other minor issue: 1) Table 2 column for "0.1 mAP", bold numbers are wrong.

Reviewer 3



Post rebuttal update: After reading the rebuttal and other reviews, I upgraded my score to 6. The clarifications should be added to the paper. ************************ Paper summary: The paper proposes an approach to estimate the 3D coordinates of objects in a scene and the room layout. The model includes three components. First, the bounding box coordinates are estimated. Second, a layout prediction module finds the floor, ceiling and walls. Third, through a physics engine the stability of the predicted scene is estimated. Paper strengths: - The paper tackles an interesting problem. Including the physics engine in the training loop is interesting (unlike previous approaches like Mottaghi et al. that use the physics engine in a passive way). - The paper provides large improvements over Factored3D, which is a state-of-the-art method. Paper weaknesses: - The paper has serious clarity issues: (1) It is not clear what "unsupervised" means in the paper. The groundtruth bounding box annotations are provided and there is physics supervision. Which part is unsupervised? Does "unsupervised" mean the lack of 3D bounding box annotations? If so, how is the top part of Fig. 2 trained? (2) It is not clear how the fine-tuning is performed. I searched through the paper and supplementary material, but I couldn't find any detail about the fine-tuning procedure. (3) The details of the REINFORCE algorithm is missing (the supplementary material doesn't add much). How many samples are drawn? What are the distributions that we sample from? How is the loss computed? etc. (4) The evaluation metric is not clear. It seems the stability score is computed by averaging the stability scores of all primitives (according to line 180). Suppose only one box is predicted correctly and the stability score is 1 for that box. Does that mean the overall stability score is 1? - Another main issue is that the results are not that impressive. The physics supervision, which is the main point of this paper, does not help much. For instance, in Table 2, we have 0.200 vs 0.206 or 0.256 vs 0.261. The fine-tuning procedure is not clear so I cannot comment on those results at this point. Rebuttal Requests: Please clarify all of the items mentioned above. I will upgrade the rating if these issues are clarified.